# Exploring the Potential of Plant-Derived Exosome-like Nanovesicle as Functional Food Components for Human Health: A Review

**DOI:** 10.3390/foods13050712

**Published:** 2024-02-26

**Authors:** Yizhi Liu, Chaoqin Ren, Ruiling Zhan, Yanan Cao, Yuanhang Ren, Liang Zou, Chuang Zhou, Lianxin Peng

**Affiliations:** 1Key Laboratory of Coarse Cereal Processing of Ministry of Agriculture and Rural Affairs, Chengdu University, Chengdu 610106, China; liuyizhi@stu.cdu.edu.cn (Y.L.); caoyanan@cdu.edu.cn (Y.C.); renyuanhang@cdu.edu.cn (Y.R.); zouliang@cdu.edu.cn (L.Z.); zhouchuang@cdu.edu.cn (C.Z.); 2College of Resources and Environment, Aba Teachers University, Wenchuan 623002, China; 18942821203@163.com; 3Agricultural Science Research Institute of Tibetan Autonomous Prefecture of Ganzi Prefecture, Kangding 626099, China; a1715127249@163.com

**Keywords:** plant-derived exosomal nanovesicles, structural composition, functional features, application potential

## Abstract

Plant-derived exosome-like nanovesicles (PELNs) are bilayer membrane-enclosed nanovesicles secreted by plant cells, serving as carriers of various substances such as proteins, RNA, and metabolites. The mounting evidence suggests that PELN plays a crucial role in transmembrane signaling, nutrient transportation, apoptosis, and regulation of gut microbiota composition. This makes it a promising “dark nutrient” for plants to modulate human physiology and pathogenesis. A comprehensive understanding of PELN formation, uptake, and functional mechanisms can offer novel insights into plant nutrition and functional properties, thereby facilitating the precise development of plant-based foods and drugs. This article provides a summary of PELN extraction and characterization, as well as absorption and delivery processes. Furthermore, it focuses on the latest discoveries and underlying physiological mechanisms of PELN’s functions while exploring future research directions.

## 1. Introduction

Extracellular vesicles (EVs) are lipid bilayer-defined particles naturally released by cells and lack functional nuclei and cannot replicate [1]. Research has shown that almost all types of cells have the ability to produce EVs [2]. Depending on their generation process, EVs can be classified into four categories: exosomes derived from the endocytic pathway, microvesicles directly released from the plasma membrane, apoptotic bodies produced through apoptosis, and tumor-derived vesicles [3]. Exosomes are subcellular vesicles characterized by a bilayer membrane structure, exhibiting a diameter ranging from approximately 40 to 160 nm [4], abundant in proteins, lipids, RNA, metabolites, and other components, mounting evidence suggests that plant-derived exosomes play pivotal roles in modulating human physiology as well as the pathogenesis of various diseases. Compared to exosomes derived from animals, plant-derived exosomes possess advantages such as diverse sources, immune tolerance, and safety. Additionally, they can serve as a natural nano-delivery system. Furthermore, exosomes have a more substantial degree of bioactivity and immunogenicity than liposomes as they are distinctly chiefly formed by cells, which improves their steadiness in the bloodstream, and enhances their absorption potential and medicinal effectiveness in vitro and in vivo [5]. Consequently, plant-derived exosomes have the potential to emerge as a prominent area of interest in food nutrition research, herbal activity evaluation, and utilization.

Although various methods have been developed for the extraction of plant-derived exosomes, it remains challenging to obtain pure exosomes due to the intricate cell structures and heterogeneity inherent in plants. Consequently, exosomes often contain other microvesicles and impurities, collectively referred to as plant-derived exosome nanovesicles (PELNs). The abundance of evidence supports the crucial involvement of PELN in intercellular signaling, defense mechanisms, and interspecies communication [6]. In recent years, research on the biological activity of PEL has been favored by scholars, including crossing the blood–brain barrier to repair alcoholic inflammation [7], improving intestinal inflammation [6,8,9,10], treating tumors [9,11,12], cardioprotection [13], remodeling intestinal microbiota [14], and promoting wound healing [15]. The PELN nanocarrier is widely recognized for its ability to safely transport active molecules to specific tissues [16]. Furthermore, through engineering modifications, its targeting and delivery efficiency have been significantly enhanced [9]. As a result, PELN exhibits immense potential for diverse applications.

Although PELN has emerged as a research focal point, numerous unresolved issues continue to impede the development and application of related products. For instance, how can active substances like miRNA and secondary metabolites in plants be selectively incorporated into PELN? How do these active substances synergistically interact within PELN? Additionally, what about the stability of PELN processing and its absorption delivery mechanism? In light of the aforementioned issues, this paper presents a comprehensive overview of the formation and extraction, absorption and delivery processes of PELN. It specifically focuses on the latest discoveries and mechanisms underlying PELN’s physiological functions, and proposes potential research directions and applications for PELN, thereby offering novel insights into the exploration and utilization of plant-derived nutrients.

## 2. Isolation and Purification of PELN

### 2.1. Formation of PELN

Eukaryotic cells generate a multivesicular body (MVB) via the endosomal sorting complex required for the transport (ESCRT) system, and there is substantial evidence supporting MVB as intracellular precursors of exosomes [17]. While most ESCRT-generated MVBs are directed to lysosomes for degradation [18], some interact with SNARE proteins at the plasma membrane [19] and are guided to the cell surface in a valve-like manner [20]. Once secreted into the extracellular environment, these MVB-derived exosomes can perform various functions [21] (Figure 1A). The PELN exhibits a concave inward morphology resembling a cup shape under electron microscopy. The study revealed that, in contrast to exosomes derived from mammalian cells, PELN exhibits selective transportation of specific plant secondary metabolites and performs distinct functions. Conducting comprehensive investigations on the utilization of PELN’s unique sorting mechanism can contribute to the production of PELN with consistent quality or enhancement of its biological activity. We will delve into this topic further in subsequent discussions.

### 2.2. Extraction Methods

The traditional methods of PELN separation are as follows: ultracentrifugation, density gradient centrifugation, precipitation method, and size exclusion chromatography.

In recent years, ultracentrifugation has become the most common method of extracting PELN because of its low price and the recognition of the purity of PELN. But under the influence of strong centrifugal force, PELN may become lost [22,23]. To address this issue, a team utilized a 30% sucrose cushion in 2015 to protect PELN. Subsequently, vesicles are purified through density gradient centrifugation using sucrose or iodixanol media based on ultracentrifugation [24]. However, density gradient centrifugation encounters challenges such as low extraction yield and complex operation.

Additionally, the combination of size exclusion chromatography (SEC) and ultrafiltration has also been documented. You et al. [16] employed conventional ultracentrifugation (UC), polyethylene glycol (PEG), and SEC+ ultrafiltration techniques for extracting ELN from cabbage, revealing comparable yields and particle sizes among the methods. However, SEC exhibited higher purity compared to PEG and UC as evidenced by the presence of multiple peaks observed during NTA analysis on ELN extracted using PEG and UC. A similar scenario was observed by Leng et al. [25] when they isolated PELN derived from blueberries. They discovered that SEC effectively reduced the formation of protein aggregates, thereby enhancing the purity. In addition, an electrophoretic-coupled separation method was employed in lemon (LELN) based on the distinct characteristics of various particle sizes and charge fluidity under the influence of an electric field. The author selectively isolated LELN by means of a 300 kDa dialysis bag, effectively eliminating other proteins and nucleic acids through the application of an electric field [26].

Recently, a team has developed a method for simplified exosome extraction through PEG precipitation [9]. Conventional polymer-based methods for exosome precipitation typically utilize PEGs with molecular weights ranging from 6000 to 20,000 Da [27]. However, whether altering the PEG precipitation conditions (e.g., pH) or adjusting the molecular weight and concentration of PEG, the resulting PELN typically exhibits a large particle size, which may account for why high levels of impurity proteins tend to polymerize with PEG. To address this issue, aqueous two-phase systems (ATPS) are employed for the optimization of the single PEG extraction method. ATPS primarily consists of polymer–polymer systems and polymer–salt systems [28]. Jin et al. [29] developed a polymer aqueous two-phase system utilizing PEG (25,000–45,000)/dextrans DEX (450,000–650,000) for the extraction of exosomes derived from animals. It was observed that traditional PEG resulted in significant protein contamination, whereas in the PEG/DEX system, due to the presence of a phospholipid bilayer in the exosomal plasma membrane and its enrichment in the lower layer DEX, most impurity proteins were dispersed within PEG. Subsequently, Kilbas et al. [30] employed Coomassie Brilliant Blue R-250 to visually validate the stratification between the PEG phases by colorizing them. The current system, however, presents certain issues. For instance, the presence of dextran in the sample leads to an increase in viscosity, resulting in uneven distribution of PELN and interference with gel electrophoresis. The impact of the dextran system on subsequent experiments necessitates attention, particularly requiring a negative control for dextran.

In summary, there is currently no flawless method for extracting PELN, and the selection of one or multiple methods should be based on specific research needs. For example, if there is a high demand for PELN yield, we can use the precipitation method to extract it. If the purity requirements are high, density gradient centrifugation and size exclusion chromatography can be used. Both are in demand, and ultracentrifugation methods can be considered. In addition, referring to traditional exosome extraction methods, such as immunoassays based on exosomal biomarker proteins, although PELN does not yet have a clear biomarker, it is a promising isolation method, such as Zarovni et al. [31] using nanomagnetic beads to immunocapture exosomes, and the results show that the yield of exosomes is 10–15 times higher than when using ultracentrifugation-based isolation techniques. Microfluidic technology is also an open and promising technology, which can be based on both physical isolation and immunoassay for exosome extraction, and overcomes some limitations faced by traditional methods, including long processing time, low purity, and risk of exosome damage [32].

### 2.3. Strategies for Enhancing the Production of PELN

The structural complexity of plant tissues distinguishes them from animal tissues, necessitating different pretreatment conditions, extraction methods, and adjustment of extraction environment for separating PELN with varying structural properties. Factors such as the impact of plant cell walls on extraction yield and the promotion of secretion in plants with low water content should be taken into consideration. The researchers in the study conducted by Suresh et al. [33] attempted to enhance the yield of GELN through PEG precipitation by manipulating the pH of the sample. This could be attributed to the potential impact of pH alteration on membrane potential, subsequently influencing ion channels and transporters’ activity and structure both inside and outside the cell membrane. Consequently, this promotes the fusion release of plasma membrane-bound multivesicular bodies (MVB) and consequently affects exosomal release efficiency.

Revising the plant growth environment to stimulate ELN release from plant cells is also a crucial approach for enhancing PELN yield. A study on Arabidopsis plants revealed that PELN levels in Arabidopsis thaliana were significantly elevated during microbial infection [34]. Therefore, it is hypothesized that biostress may enhance the yield of PELN. However, due to the paucity of relevant research, pertinent evidence can only be extrapolated from studies conducted on mammals. Liu et al. [35] reported an increase in exosome levels following a high-fat diet (HFD) and fatty acid (FA) intake. Additionally, several other investigations have demonstrated augmented exosome secretion during biological stress [36,37,38]. Hence, it is reasonable to speculate that the pathways of PELN and mammalian exosome secretion exhibit similarities, with an increase in secretion observed during periods of biological stress [39]. This phenomenon may be closely associated with the lysosomal pathway, which has a huge impact on the release of PELN. Zhang et al. [40] discovered that damage regulatory autophagy modulators (DRAMs) are among the genes that promote exosome secretion in NAFLD patients, and they found that DRAM may be induced by altering lysosomal permeability to increase levels of HFD- and FA-induced exosomes. While lysosomes play a crucial role in the body, inhibiting their function appears to be a promising approach for enhancing PELN levels in plants.

### 2.4. Characterization of PELN

The characterization of PELN is often accomplished through a combination of methods as supplementary means of characterization, particularly in terms of its visual appearance, shape, and size. Nanoparticle tracking analysis (NTA) is a high-throughput visualization technique that utilizes laser beam monitoring of Brownian motion in liquid suspensions to determine particle size and concentration [41]. The major advantage of NTA is its rapidity and low sample requirements, enabling effective monitoring of particles below 200 nm. However, it should be noted that NTA necessitates high sample purity and may not be suitable for analyzing more dispersed particle systems [42,43,44]. Therefore, in practical applications, following the determination of PELN particle size and concentration through NTA analysis, single vesicles are commonly characterized using scanning electron microscopy (SEM) and atomic force microscopy (AFM). The AFM technique utilizes scanning probes to interact with exosomes on the surface molecules, enabling observation of the sample surface topography. The key distinction between AFM and SEM lies in replacing the tunnel probe with a sharp needle tip that is fixed at one end and mounted on an elastic microcantilever at the other end. Additionally, instead of detecting the tiny tunnel current, AFM detects the minute deformation generated by the force exerted on the microcantilever [45]. The use of AFM and SEM enables the acquisition of high-resolution images depicting the structure of exosomes. However, it should be noted that sample preparation for SEM has the potential to alter exosome morphology, while the electron beam employed in these techniques can cause damage to exosome [46]. Additionally, dynamic light scattering (DLS), flow cytometry (FACS), and ZETA potential analysis have also been employed for the characterization of PELN. Furthermore, apart from visual observation of PELN’s appearance, shape, and size, integrating intuitive data for assessing the quality of PELN is typically necessary. However, the characterization of PELN purity poses a challenge as NTA lacks specificity in distinguishing PELN from other particles. SEM is capable of distinguishing PELN from other particles, but it lacks the ability to quantitatively measure soluble contaminants in a given sample. As a result, the MISEV2018 guideline suggests employing semi-quantitative approaches such as evaluating the granule/protein, granule/lipid, and lipid/protein ratios for characterizing the purity of extracellular vesicles [1] However, the characterization of PELN purity using this method remains limited. You et al. [16] extracted PELN from cabbage and achieved a purity level of only 10^10^, which is clearly inadequate compared to the clinical treatment standard of 10^11^ found in mammalian sources. As research on PELN progresses, achieving clinical therapeutic levels of purity becomes essential, although we still lack sufficient evidence regarding the optimal level.

## 3. Composition of PELN

The main constituents of PELN are proteins, lipids, nucleic acids, and phytochemicals specific to different plants. In this section, we will analyze the similarities and differences in the components derived from various sources of PELN and discuss their respective roles in the biological composition of PELN.

### 3.1. Protein and Lipids

PELN has discussed the basic structure of PELN lipids and proteins in several reviews, for example, PELN proteins are mainly composed of stimulus, defense, annexin, etc., lipids are mainly composed of phospholipids and glycolipids, phosphatidylcholine (PC), phosphatidic acid (PA), phosphatidylethanolamine (PE), phosphatidylinositol (PI) and phosphatidylglycerol (PG), digalactosyldiacylglycerol (DGDG), monogalactosyldiacylglycerol (MGDG), and monogalactosyl monoacylglycerol (MGMG), among others [21,47]. Here, we focus on the potential of protein as a key factor in yield enhancement and as a biomarker, as well as the relationship between lipid diversity and functional differences.

Limited studies have shown that [48] PELN proteins may be involved in cross-border communication and signaling and that PELN may contain a high abundance of stimulated proteins, which may contribute to the increase in stress yield or to study the differential properties of stress function [34]. In the past, the quadruple transmembrane proteins CD9, CD63, and CD81 were generally recognized as universal markers of exosomes due to their accumulation in small EVs and the homeostatic accumulation of CD63 in MVBs. However, Kowal et al. [49] found that CD9 and CD81 were also detected in non-exosomal impurities. A similar recent study found that exosomes could not be formed in the presence of CD9 and CD81 without CD63. However, in the absence of CD9 and CD81, CD63 and one or two other quadruple transmembrane proteins may also facilitate the formation of new exosomes [50]. Therefore, CD63 may be a reliable target for universal markers of exosomes. Recently, in the field of edible plants, exosomal transmembrane or lipid-binding proteins such as CD9, CD63, CD81, and cytosolic protein TSG101 have been found in bitter melon and grapefruit [12]. CD9, CD81, and TSG101 should be considered with caution in view of the controversy over universal markers in exosomes. CD63 is currently accepted as a positive protein marker for animal-derived exosomes, so CD63 provides a new idea for the characterization of PELN. However, although Hoshino et al. [51] verified the feasibility of CD63 as a universal marker for exosomes in mammals with a large number of samples, there is still a lot of experimental evidence to be demonstrated whether it can be included as one of the universal markers of PELN.

Lipids are an important component of exosomes, and in mammalian exosomes, the function of lipids has been studied in depth, mainly including participation in exosome formation and release, immune regulation, targeting, signal transmission, etc. This may be related to the diversified lipid composition, for example, phosphatidic acid (PA) is one of the most important components of the cytoplasmic membrane, which is produced by phospholipase D (PLD) and PLC/DGK with PC, PE, and PG as substrates. It is worth noting that PA is closely related to the anti-stress response of plants, and when plant cells are exposed to microbial pathogens or inducers, PLD is activated, and PA is rapidly produced through the mechanism of ABA signaling cascade, thus participating in defense signaling [52]. In addition, the membrane structure of PELN includes phosphatidylcholine (lecithin PC) and phosphatidylethanolamine (cerebral phospholipid PE), and the relative proportions of PELN from different sources are different, which may be an important reason for the targeting characteristics of different PELNs, such as grapefruit PELN with higher PE content migrating to the liver through the intestine, while ginger PELN with higher PA content accumulates in the intestine [14,53].

There is another major class of lipids—glycolipids in the cytoplasmic membrane derived from plants, mainly digalactosyldiacylglycerol (DGDG), monogalactosyldiacylglycerol (MGDG), monogalactosylmonoacylglycerol (MGMG), etc. Ginger contains high levels of DGDG, and MGDG [14]. It has been reported that DGDG and MGDG appear to be one of the reasons for the stability of PELN, and it has also been found that DGDG derived from oat PEL can affect the exosome pathway of dectin-1 through the hippocampal calcitin (HPAC)-β-glucan pathway [7] (dectin-1 is a β-dextran pattern recognition receptor that plays an important role in antifungal immunity [54]).

### 3.2. Nucleic Acids

Exosomes contain a plethora of non-coding RNAs, including microRNAs, lncRNAs, circRNAs, and siRNAs, which play a crucial role in intercellular communication. The proportion and diversity of RNA species in PELN exosomes were lower than those found in animal-derived exosomes, with miRNA being the predominant type [55].

MiRNAs are highly conserved non-coding small RNA molecules, typically ranging in length from 18 to 22 nucleotides. They play a crucial role in guiding the RNA-induced silencing complex (RISC) to degrade or inhibit the translation of target gene mRNA by forming base pairs with specific regions. The expression of these miRNAs is tightly regulated and restricted to specific tissues and developmental stages due to their remarkable conservation, temporal specificity, and tissue-specific characteristics [56]. However, it is worth noting that the miRNA species identified in plants may not always align precisely with those found in PELN, implying the involvement of sorting mechanisms for miRNA regulation. For example, Liu et al. [57] found that Tartary buckwheat ELN (TELN) carries 29 unique miRNA families and 11 new miRNAs compared to Tartary buckwheat seeds. In addition, Teng et al. also found that the miRNA of ginger-derived ELN differs in structure from the miRNA structure of ginger tissue. Garcia-Martin et al. [58] revealed that the preference for specific bases could potentially serve as a mechanism for sorting miRNAs within exosomes. Additionally, Leidal et al. [59] discovered that the autophagy protein LC3 coupling mechanism governs the sequestration of miRNA by PELN along its pathway. In essence, the loading of cargo onto PELN is not solely determined by the characteristics of the cargo itself but also influenced by the structure of PELN.

The relationship between endogenous miRNAs in hosts and diseases is relatively well established, whereas the effects of plant-derived miRNAs on the body remain controversial, mainly focusing on their stability and effective concentration. Previous studies have demonstrated the stability of an atypical miRNA-miR2911 in honeysuckle decoction soups [60]. Additionally, Chapado et al. [61] also showed that the main miRNAs derived from broccoli are relatively stable in food processing engineering, but the processing and digestion stability of most plant miRNAs has not been fully demonstrated. The latest research findings indicate that plant miRNAs are encapsulated within exosomes, enabling them to evade degradation by gastrogut digestive enzymes. However, the stability of various types of PLEN in the gastrogut tract remains uncertain [62]. Additionally, the regulation of transboundary parasites by PELN-loaded miRNA has also been documented. Qiang et al. [63] demonstrated that miRNAs derived from Arabidopsis ELN can effectively target Botrytis cells through vesicle transport system-mediated loading and action, resulting in the downregulation of the Botrytis virus gene and subsequent reduction in infectivity.

Although there are still numerous considerations in the study of plant-derived miRNA, mounting evidence suggests that miRNAs possess transboundary regulatory functions. In a recent honey ELN investigation, it was discovered that miRNA4057 downregulates the expression of cytokines IL-6 and TNF-α through specific pathways [8]. In the study of ginger ELN, gma-miR396e inhibited colitis by regulating the expression of the *Lactobacillus rhamnosus* fimbillus gene (spac), preventing LGG aggregation in the gut mucosa [14]. Another prediction of miRNA target functionality in blueberry ELN found that miR-156e, miR-162, and miR-319d have potential targets PIGIs, MAPK, and PDE7A [57]. These are all associated with inflammation, suggesting that the miRNA loaded with blueberry ELN may play a regulatory role in inflammation-mediated related abnormal responses.

However, a plant PELN harbors multiple miRNAs, with the potential for one miRNA to target multiple genes and for a gene to be regulated by multiple miRNAs. The utilization of bioinformatics in target prediction serves to narrow down the scope of research. For instance, Chapado et al. [61] utilized bioinformatics to predict the targeting properties and functionality of exogenous miRNAs derived from broccoli on human genes, suggesting that miRNAs may play a role in the functional mechanism of broccoli; Teng et al. [14] employed bioinformatics tools to identify mdo-miR7267-3p as a highly expressed miRNA in ginger-derived ELN that targets the ycne gene, and further experimentally confirmed that upregulation of this gene can enhance the production of anti-inflammatory factor IL-22 through AHR pathway, thereby ameliorating gut inflammation. The binding site of miRNA in Tartary buckwheat exosomes to *Lactobacillus rhamnosus* was predicted using bioinformatics by Liu et al. [57], and subsequently validated through in vitro experiments. Bioinformatic analysis showed that miRNA in TELN had binding sites with the LGG genome, but in vitro experiment results showed that miRNA proliferation to LGG significantly decreased without vesicle protection. It is worth noting that the miRNAs contained in the PELN often exhibit synergistic effects. Moreover, their targeting specificity is relatively weak, which poses a significant limitation to their comprehensive investigation and clinical application.

### 3.3. Other Phytochemical Components

As PELN is derived from pinocytosis, it is expected that a wide range of highly specific phytochemicals present in the intercellular fluid and cytoplasmic matrix will be carried and potentially enriched during the process. For instance, an ELN (ONV) obtained from orange juice demonstrates enrichment in leucine, threonine, formate, methanol, ethanol, and sn-glycerol-3-phosphocholine compared to orange juice [64]. The study revealed that ginsenoside Rg3 exerts a significant effect on restraining tumor cell proliferation and suppressing tumor neovascularization, thus rendering it a promising candidate for anti-tumor therapy [65]. While conducting research on ginseng-derived extracellular matrix-like nanoparticles (GELN), Cao et al. [66] discovered that GELN exhibited an enrichment of ginsenoside Rg3. Subsequent pharmacological experiments revealed that the high concentration of ginsenoside Rg3 in GELN significantly suppressed melanoma growth in tumor-bearing mice, suggesting a potential correlation between this enrichment and its inhibitory effects. Additionally, Zhuang et al. [9] discovered that GELN also significantly enriched a substantial amount of 6-gingerol. Through lipid knock-in and knock-out strategies, it was revealed that 6-gingerol plays a crucial role in the prevention of alcohol-induced liver injury via the TLR3/TRIF pathway. However, PELN exhibited specific enrichment behavior for aloin and aloe-emodin, which were found to have higher content in aloe vera. On the other hand, β-sitosterol (a stabilizer used to enhance liposome stability by promoting ordered phospholipid arrangement) showed higher content in aloe vera’s PELN [67].

Woith et al. [68] propose that secondary metabolites in citrus fruits are predominantly packaged into PELN through passive transport, as lipophilic secondary metabolites tend to be more enriched in PELN. However, this phenomenon is not absolute; for instance, PELN derived from grapefruit has been found to carry naringenin, whereas the presence of naringenin in PELN derived from oranges has not yet been detected [64,69].

Traditionally, the nutritional or medicinal value of purified plant secondary metabolites has been assessed through their activity. However, it can be speculated that encapsulation of the active ingredient in PLEN may alter its stability, bioavailability, metabolic pathway and other factors. For instance, the gut aggregation of pure β-glucan poses a challenge for its transportation across the blood–brain barrier; however, Xu et al. [7] demonstrated that β-glucan present in oat-derived ELN can effectively traverse the blood–brain barrier to attenuate brain inflammation and enhance cognitive function in alcohol-fed mice. In addition, as a tumor treatment drug, methotrexate (MTX) has certain side effects. Wang et al. [69] coupled grapefruit-derived PELN with MTX to greatly reduce its toxicity without compromising efficacy. Zhang et al. [6] discovered that ginger exosomes (GDNPs) are primarily absorbed by gut epithelial cells (IECs) and macrophages, which can alleviate acute colitis, enhance gut repair, and prevent chronic colitis and colitis-related cancers (CAC); GDNPs1, GDNPs2, and GDNPs3 were obtained by selecting exosomes from different sites in the density gradient. The content of 6-gingerol and 6-shogaol was compared between GDNPs2 and lower in GDNPs1. Furthermore, oral administration of GDNPs2 increased the survival and proliferation of IECs, reduced pro-inflammatory cytokine expression (such as TNF-α, IL-6, and IL-1β), and increased anti-inflammatory cytokine expression such as IL-10 and IL-22. The findings suggest that GDNPs2 possess the ability to inhibit factors detrimental to the gut while promoting gut repair. Consequently, it is postulated that 6-gingerol and 6-shogaol present in GDNPs may contribute to the anti-inflammatory functionality of GDNPs2, with exosome delivery being implicated in its realization. Sun et al. [70] demonstrated a novel approach utilizing exosomes for targeted delivery of anti-inflammatory agents, such as curcumin, to activated myelocytes in vivo. The use of exosomes resulted in enhanced stability and concentration of curcumin in the bloodstream. Their findings highlight the role of exosomes in determining target specificity and improving the anti-inflammatory activity of curcumin by facilitating its uptake into inflammatory cells. Notably, this enhancement is attributed to the involvement of exosomes rather than lipids alone.

## 4. Biological Function of PELN

The biological function of PELN has garnered increasing attention, given its richness in a variety of active molecules and the fact that research on its mechanism of action is still in its nascent stages. We have compiled the primary mechanistic studies conducted on the biological functional characteristics of PELN in recent years (Table 1).

### 4.1. Absorption and Distribution of PELN in the Body

Plants serve as a vital source of food and medicine for humans, and their exosomes can enter the body through various pathways. Aimaletdinov et al. [92] conducted a systematic analysis of EV distribution in animals with different dietary patterns. The mode of administration and the composition of extracellular vesicles (EVs) are critical determinants for their biodistribution. Similar observations have been reported for plant-derived EVs, where ginseng-derived vesicles were found to accumulate primarily in the liver and spleen following intravenous or intraperitoneal injection, while oral administration resulted in preferential distribution within the gastrogut tract [66]. This may be attributed to direct transport via systemic circulation with injections versus recirculation through digestive barriers with oral delivery. The oral route is the most commonly used administration method, and there are significant variations in the distribution of PELN within the body following oral administration from different sources. For instance, Berger et al. [64] reported that PELN derived from orange juice is exclusively present in the intestine, while Zhuang et al. [80] observed that a significant proportion of ginger-derived PELN accumulates in the liver and exhibits a prolonged half-life of over 12 h. These examples illustrate that the distribution of PELN is influenced by its structural composition [14], size, charge [93] and other factors, as well as its ability to traverse gut barriers. However, there is still a dearth of systematic research on the in vivo distribution behavior and mechanism of PELN, and the relevant reports on human vesicles can serve as a valuable reference [51]. Furthermore, nasal administration of oat-derived PELN (OELN) has demonstrated its ability to traverse the blood–brain barrier and be absorbed by microglia through the β-glucan-HPAC hippocampal calcin pathway, thereby exerting biofunctional activity [7]. Additionally, garlic-derived PELN appears to be assimilated by brain microglia, suggesting that apart from crossing the gut barrier successfully, PELN may also effectively penetrate the blood–brain barrier [76].

### 4.2. Reconstructing Gut Microbiota

Gut microbiota are intricately associated with the body’s immune response, nutritional uptake, metabolic processes, and other physiological functions. They play a pivotal role in regulating energy metabolism, maintaining gut barrier function, and activating mucosal immunity [94]. Targeting gut microbiota through dietary interventions represents a potent approach to achieving personalized nutrition. While research has predominantly focused on exploiting plant exosomes as delivery systems for targeted drug development to cells, there is a dearth of comprehensive investigation into the delivery and subsequent response of gut microbiota [95]. The targeted delivery capabilities of PELN have the potential to precisely regulate the composition of gut microbiota. The study revealed that PELN was capable of influencing the abundance of *Bacteroides* and *Lactobacillus* in the gut microbiota. The study conducted by Teng et al. [14] revealed, based on 16sRNA analysis, that ginger-derived ELN is able to be absorbed by the gut microbiota, and ginger-derived GELN has the potential to enhance the abundance of *Lactobacillus* and *Bacteroides* while reducing the prevalence of *Clostridium*. More reliably, their clinical data also support such a conclusion. It is well known that PELN carries miRNAs that regulate gene expression, and they screened out a common probiotic, *Lactobacillus rhamnosus* (LGG), and through further bioinformatics analysis, the mdo-miR-7267-3P carried by PELN increased tryptophan-I3A of LGG by promoting the expression of single oxygenase ycnE, and I3A increased the expression of the anti-inflammatory factor IL-22 through the AHR pathway. In addition, they also found that the ath-miR167a-5p carried by PELN downregulated the expression of the LGG fimbriae gene spac, thereby preventing the accumulation of LGG, further elucidating the potential role of GELN-LGG in inflammatory bowel disease. Another study of lemon-derived nanovesicles (LELN) also found that LELN promoted the growth of *Lactobacillus enterotica* [75].

Liu et al. [57] found through bioinformatics tools that some miRNAs in PELN derived from Tartary buckwheat can target functional genes in the physiological processes related to *Escherichia coli* and *Lactobacillus rhamnosus*, thereby significantly promoting the growth of *Escherichia coli* and *Lactobacillus rhamnosus*, and by co-fermenting Tartary buckwheat exosomes with fecal microorganisms, it was found that Tartary buckwheat exosomes can change the composition of gut microbiota and promote the production of short-chain fatty acids. Currently, the understanding of the interaction between PELN and gut microbiota is limited. Further research is needed to investigate how PELN is selectively absorbed by gut microbiota and delivers active molecules to specific microorganisms in order to intervene in the mechanism of action of gut microbiota. The use of bioinformatics tools can help narrow down the scope of research, which will facilitate a better understanding of the mechanism behind PELN’s actions on gut microbiota.

### 4.3. Ameliorating Inflammation

Inflammatory bowel disease (IBD) is a chronic inflammatory disorder of the gastrogut tract, encompassing ulcerative colitis (UC), Crohn’s disease (CD), and undetermined colitis (IC). As the primary point of entry for dietary components into the body, Peyer’s patches in gut-associated lymphoid tissue play a crucial role in mitigating gut inflammation through diverse mechanisms (Figure 2B). Ju et al. [9] administered grape-derived nanovesicles (GELN) via gavage and observed that GELN facilitated the self-renewal of gut stem cells through the Wnt/β-catenin pathway, which is known to suppress inflammatory bowel disease. Specifically, Lgr5-EGFPhi gut stem cells internalize GELN via pinocytosis, facilitating the entry of GELN into the cell and subsequent nuclear accumulation of β-catenin, thereby initiating activation of the Wnt signaling pathway. This subsequently triggers the Tcf4 transcriptional mechanism, resulting in the upregulation of a cascade of growth-promoting genes such as Sox2, Nanog, OCT4, KLF4, c-Myc and EGFR. Zhang et al. [6] demonstrated that GELNs exhibited a preferential affinity for the mouse colon and effectively stimulated the proliferation of gut epithelial cells through oral administration of GELN. The administration of GELN was found to result in reductions in apolipoprotein (Lcn-2) levels in mice with DSS-induced IBD, accompanied by a decrease in spleen weight and an increase in colon length. Further protein analysis revealed a significant reduction in colonic myeloperoxidase (MPO), an indicator of neutrophil infiltration, while E-cadherin, which plays a crucial role in epithelial cell–cell adhesion and tissue structure maintenance, exhibited a significant increase. The authors’ conclusion was further supported by the observed changes in inflammatory factors, which provided additional evidence for the beneficial effects of GELN on IBD. Specifically, GELN was found to enhance the proliferation of gut epithelial cells and mitigate detrimental factors. In a study conducted by Deng et al. [10], it was discovered that nanovesicles derived from broccoli are involved in the upregulation of AMPK protease expression, resulting in reduced phosphorylation due to downregulation of rapamycin target (MTOR)/S6 kinase (S6k). This ultimately leads to increased tolerance of gut cells towards dendritic cells (DCs). By characterizing inflammatory factors such as IFN-g, interleukin, and TNF-α, they observed that enhanced tolerance towards dendritic cells effectively suppresses colitis.

Additionally, Zhuang et al. [80] discovered that 6-gingerol in GELN induces Nrf2 nuclear translocation through modulation of the TLR4/TRIF pathway, leading to dissociation from Keap1 and subsequent translocation into the nucleus. This results in the formation of a heterodimer with Maf and further activation of ARE-mediated downstream gene expression, ultimately promoting the repair process for alcoholic liver injury (Figure 2D). The GELN compound has demonstrated the ability to specifically target NLRP3 inflammasomes, leading to a significant improvement in inflammation [73]. Similar findings have been reported in the HELN study on honey sources. Although HELN does not affect the levels of sensory protein NLRP3, adaptor protein ASC, and enzyme procaspase-1, it appears to inhibit the oligomerization of ASC adaptor protein, thereby preventing the formation of the NLRP3 inflammasome platform [8]. Liu et al. [78] identified a similar mechanism in shiitake mushrooms, while De Robertis et al. [79] utilized high-throughput sequencing of blueberry-derived nanovesicles combined with bioinformatics to predict the target genes of three miRNAs and found that they may regulate ROS levels through the TNF-α pathway (Figure 2A). Additionally, the mechanism of PELN’s action on brain inflammation has also attracted scholarly attention. Xu et al. [7] discovered that OELN can be internalized by cells through the β-glucan-hippocampal calcin (HPAC) pathway, while OLEN’s DGDG inhibits brain inflammation by blocking the activity of β-glucan and dectin-1, thereby facilitating its internalization and recycling without being directed towards the lysosomal pathway. Moreover, Sundaram et al. [76] found that garlic-derived nanovesicles (GELN) can be taken up by microglia via PA-mediated transmembrane action and interact with brain acid soluble protein 1 to form a competitive inhibitor for CaM binding, consequently suppressing c-MYC expression and ameliorating LPS-induced brain inflammation (Figure 2C). In addition, Teng et al. [81] found that miRNAs carried by ginger-derived ELN can inhibit the key proteins Nsp12 and Nsp13 of new coronary pneumonia.

### 4.4. Tumor Inhibition

The primary challenge in utilizing PELN as a potential therapeutic modality for cross-regional regulation lies in assessing its cytotoxicity, which necessitates an initial evaluation of its impact on normal cellular physiological activity and subsequent examination of its ability to inhibit abnormal cells. Extensive experimentation has demonstrated that PELN derived from cabbage, blueberry, strawberry, grapefruit, bitter melon, carrot, camellia, oats, and other sources does not exert any adverse effects on normal cell lines such as HaCaT, A375, HDF, and ADMSC; instead, it promotes cell proliferation. Moreover, PENL exhibits inhibitory properties against abnormal cell lines including A549, S W480, and LAMA84 [11].

Studies have demonstrated that diverse sources of PELN exert a favorable impact on human tumor cells through distinct mechanisms of action. For instance, ELN derived from bitter melon effectively suppresses the invasion and migration of the U251 human glioma cell line by modulating the P13K/AKT pathway [84] (Figure 3A). Furthermore, ginseng-derived ELN impedes the migration, invasion, cloning, and adhesion abilities of A549 and H1299 lung cancer cell lines by downregulating thymidine phosphorylase mRNA expression in the pentose phosphate pathway [86] (Figure 3B). Moreover, in studies investigating colon cancer, lemon-derived ELN has been shown to inhibit colon cancer through distinct mechanisms. Raimondo et al. [83] demonstrated that LELN suppresses cancer cell growth by attenuating ERK/p38-MAPK phosphorylation levels, thereby impairing signaling pathways. In addition, Raimondo et al. [11] revealed that LELN induces the Caspase cascade via the TRAIL pathway, ultimately mediating apoptosis in colon cancer cells (Figure 3C). Furthermore, in multiple melanoma studies, ELN derived from grapefruit exhibited a reduction in cancer cell viability by downregulating the phosphorylation level of ERK/AKT, thereby impairing its signaling [12], Cao et al. [66] discovered that ginseng-derived nanovesicles (GELN) promoted M2-to-M1 polarization through the TLR-4/MyD88 pathway. They demonstrated that this pathway is associated with GELN’s lipids or proteins rather than nucleic acids or other contents (Figure 3D).

In addition, PELN also has great potential for protecting the heart [13], promoting the expression of beneficial genes in the skin [96], and promoting placental growth and activity [91], although the mechanism has not yet been revealed (Table 1).

## 5. Potential Application of PELN

### 5.1. Stability of PELN

The stability of PELN significantly impacts the accuracy of subsequent analysis and the development and utilization of the product. Numerous studies have examined the robustness of PELN under diverse conditions, including variations in temperature, time, freeze–thaw cycles, and external physical treatments. Kim et al. [97] investigated the stability of PELN at different temperatures (−20 °C, 4 °C, 25 °C, 45 °C) and durations, revealing a decline in pH, size reduction, and decreased total protein content with increasing temperature. Over time, there is a decrease in total protein levels; however, vesicle size appears to increase during storage. In a study on camellias, it was found that repeated freeze–thaw cycles led to an increase in vesicle size and a decrease in surface potential [71]. Zhang et al. [6] demonstrated that both GDNP 1 and GDNP 2 exhibit tolerance towards freeze/thaw cycles and remain stable at room temperature, while GDNP 3 is unstable. In short, the stability of PELN is closely related to its structural composition. The impact of temperature and freeze–thaw cycles on PELN is evident, leading to the widely recognized storage environment of −80 °C. However, this may still result in changes to the exosome’s morphological structure [98]. Charoenviriyakul et al. [99] divided B16BL6 melanoma exosomes into three parts for comparison of morphology, protein content, and pharmacokinetic effects after storage at −80 °C, freeze-drying with trehalose, or direct freeze-drying. The results demonstrate that the addition of trehalose effectively prevents exosome aggregation during lyophilization, enhances their colloidal stability, and maintains their morphology. Lyophilized exosomes stored at room temperature exhibit similar protein content to those stored at −80 °C. Therefore, we recommend maintaining a low temperature environment (4 °C) throughout the PELN pre-treatment process and vacuum freeze-drying the extracted PELN for storage at −80 °C. Additionally, adding some trehalose as a cryoprotectant before freeze-drying can prevent pressure damage to the structure of PELN during lyophilization.

The effective execution of PELN’s biological function in product development necessitates the ability to withstand the challenging conditions posed by various processing environments and gastrogut digestive juices. Extensive research has demonstrated that PELN exhibits remarkable resistance against gastrogut digestion in both simulated settings and mouse models [26,69]. As previously mentioned, most PELNs can be distributed to various tissues in the body after oral administration, indicating their resistance to gastrogut digestion. While the stability of PELN in fruits and vegetables processed using non-thermal methods remains intact, its preservation in most thermally processed foods is rarely reported, possibly due to the intricate process of isolating and identifying exosomes as well as the challenges associated with studying changes induced by thermal processing. Although Ly’s review [48] concluded that the activity of PELN is susceptible to boiling and sonication, future research is needed to elucidate variations in PELN stability across different structures and plant matrices, while identifying appropriate processing methods for preserving exosome stability.

### 5.2. Security of PELN

Current research indicates that PELN derived from natural plants exhibits no toxicity towards normal cells and possesses certain cell proliferation functions. These findings have been reported across various sources of PELN, including cabbage, blueberry, strawberry, grapefruit, bitter melon, carrot, ginger and others. Furthermore, studies have demonstrated the efficacy of PELN in treating abnormal cells such as H_2_O_2_-induced MDSC cells [100], STS-induced HaCaT and HDF cells [16], with a significant pro-apoptotic effect.

It should be noted that while there is currently limited research on the safety of plant exosome particles in humans, this does not imply that all PELNs are safe. Most of the literature reports focus on traditional fruits and vegetables, which have high safety profiles and emphasize the nutritional functional characteristics of PELN. However, many plant-derived herbs and foods exhibit certain levels of toxicity or allergenicity; thus, it remains unclear whether these adverse effects are related to their PELN content. This topic warrants further investigation as it can provide a more comprehensive perspective on the quality evaluation of herbal or plant-based foods.

### 5.3. Endocytic Uptake and Intracellular Internalization of PELN

To be functional, PELN necessitates in vivo uptake and cellular internalization. Currently, researchers primarily investigate the in vivo uptake pathway of PELN through visualization techniques such as fluorescent labeling, electron microscopy, or laser confocal methods. For instance, Teng et al. [14] employed infrared fluorescent membrane dye (DiR) to label ginger-derived ELN and observed its preferential accumulation in the intestine, whereas grapefruit-derived ELN exhibited a higher tendency for liver accumulation. Mu et al. [72] utilized confocal analysis to demonstrate that four edible-plant-derived PELNs accumulate in the gut. Zhuang et al. [80], employing DiR combined with confocal analysis, identified ginger-derived PELN presence in the liver. Kumaran Sundaram et al. [76] detected a cerebral signal of garlic-derived ELN, while Teng et al. [81] observed multiple signals of ginger-derived ELN in the lungs. Although these reports indicate distinct distributions of PELN from different plant sources within an organism, the specific mechanism underlying their distribution remains unclear.

The intracellular localization of PELN underscores the significance of investigating its internalization mode in elucidating its distribution mechanism. Previous studies have identified three types of pinocytosis, namely clathrin-mediated pinocytosis, caveolae-mediated pinocytosis, and macropinocytosis [101].

Macropinocytosis is a form of endocytosis that involves highly folded regions/protrusions of the plasma membrane on a larger scale. The binding of cell relaxin D to actin nucleation and F-actin growth ends inhibits the polymerization reaction, making it commonly used for macropinocytosis suppression. Wang et al. [69] demonstrated that grapefruit-derived PELN was internalized by macrophages through macropinocytosis upon treatment with cytorelaxin D. Ju et al. [9] reported a similar finding, showing the internalization of grape-derived PELN by gut stem cells via macropinocytosis. Chlorpromazine, known for its ability to inhibit clathrin-mediated endocytosis, induced the assembly of clathrin networks on the endosomal membrane and prevented the formation of membrane fovelets on the cell surface. Exploiting this inhibitory effect, Wang et al. [69] revealed that grapefruit-derived PELN was internalized by macrophages through a clathrin-mediated pathway.

In addition to the inherent targeting ability of PELN from various sources, engineering modifications offer a promising approach (Table 2). Wang et al. [53] demonstrated that the incorporation of grapefruit-derived ELN enhanced the antiproliferative effect of methotrexate (MTX) on macrophages, while effectively mitigating MTX-induced toxicity. Zhuang et al. [80] developed an effective and safe tumor-targeting system by modifying PELN with folic acid (FA)/polyethyleneimine (PEI)/GNV (grapefruit-derived PELN). Folic acid serves as a highly efficient tumor-localizing agent due to its high-affinity binding to folate receptors that are overexpressed on many human tumors but not on non-tumor cells. PEI enhances nucleic acid delivery efficiency, although it is toxic; however, GNV’s natural encapsulation system significantly reduces PEI toxicity. Xiao et al. [102] discovered that the lemon-derived ELN, loaded with doxorubicin, contributes to cancer drug resistance. Previous studies have discussed engineering modifications for PELN in detail [103], which will not be further elaborated in this article.

## 6. Prospects

Although numerous studies have demonstrated the involvement of PELN in various crucial physiological processes, such as immune regulation, intercellular communication, and disease pathogenesis, several challenges still exist in this research field. On one hand, the isolation and purification of plant-derived exosome particles pose a significant obstacle due to the presence of plant cell walls. Despite density gradient centrifugation being considered the gold standard for extracting plant exosomes, it remains challenging to eliminate impurities and minimize potential impacts on PELN activity during extraction procedures. Additionally, issues such as expensive equipment requirements, complex operating protocols and low recovery rates further complicate this process. Therefore, there is a need for a breakthrough in the development of a simple, high-purity, high-recovery and low-damage method for extracting PELN. To address this challenge, future experimental designs could incorporate novel filtration techniques, microfluidic devices, or the application of nanotechnology to enhance the isolation process. Collaborations with experts in materials science and chemical engineering could lead to the development of new materials or surfaces specifically tailored for the selective binding and release of PELN.

Additionally, the complex mechanism behind how active molecules are sorted into plant exosomes must be understood to obtain stable and high-quality PELN. However, the mechanism by which molecules are sorted into exosomes or retained in cells remains largely unknown. Ruben Garcia-Martin et al. [56] demonstrate that miRNAs possess sorting sequences that determine their secretion in small extracellular vesicles (Sev) or cellular retention and that different cell types make preferential use of specific sorting sequences, thus defining the sEV miRNA profile of that cell type. The findings of this study offer valuable insights into the correlation between circulating exosomal miRNAs and tissues, thereby serving as a potential reference for future research in PELN. Advanced molecular biology techniques such as CRISPR-Cas9 gene editing could be employed to manipulate the expression of candidate genes involved in the sorting process. Future research should also focus on investigating the differences in PELN composition under varying environmental conditions. Furthermore, more attention needs to be paid to understanding the function of proteins, lipids and secondary metabolites in PELN as well as their synergistic relationship with RNAs. As such, further input from scholars is necessary to fully realize the potential applications of PELN.

PELN exhibits immense potential in the domains of food, medicine, and even pesticides. In the realm of food research, PELN possesses the capability to emerge as a novel criterion for assessing nutritional quality, thereby revolutionizing food processing techniques. Furthermore, it can be employed for formulating innovative food additives aimed at enhancing nutritional value. Within pharmaceutical research, PELN has also garnered significant attention as a promising candidate for investigating botanical drug activity. While traditional herbal medicine research primarily focuses on secondary metabolites’ biological activities, numerous plant secondary metabolites encounter challenges such as limited bioavailability and instability. The theory of gut microbiota proposes that these secondary metabolites undergo a transformation in the intestine to enhance their bioavailability and stability. Natural PELN contains specific secondary metabolites, whose pathways and metabolic processes are likely to be altered, thereby offering a new avenue for investigating the mechanism of action of botanicals. Additionally, vesicles present in PELN can serve as targeted delivery platforms for nutrients or drugs to improve human health. In terms of agricultural applications, PELN is closely linked with plant growth and quality formation; hence, improving plant growth and quality represents a novel research direction for PELN. Furthermore, PELN may also be utilized as an innovative biopesticide for bacteriostasis or enhancing biostress resistance.

## Figures and Tables

**Figure 1 foods-13-00712-f001:**
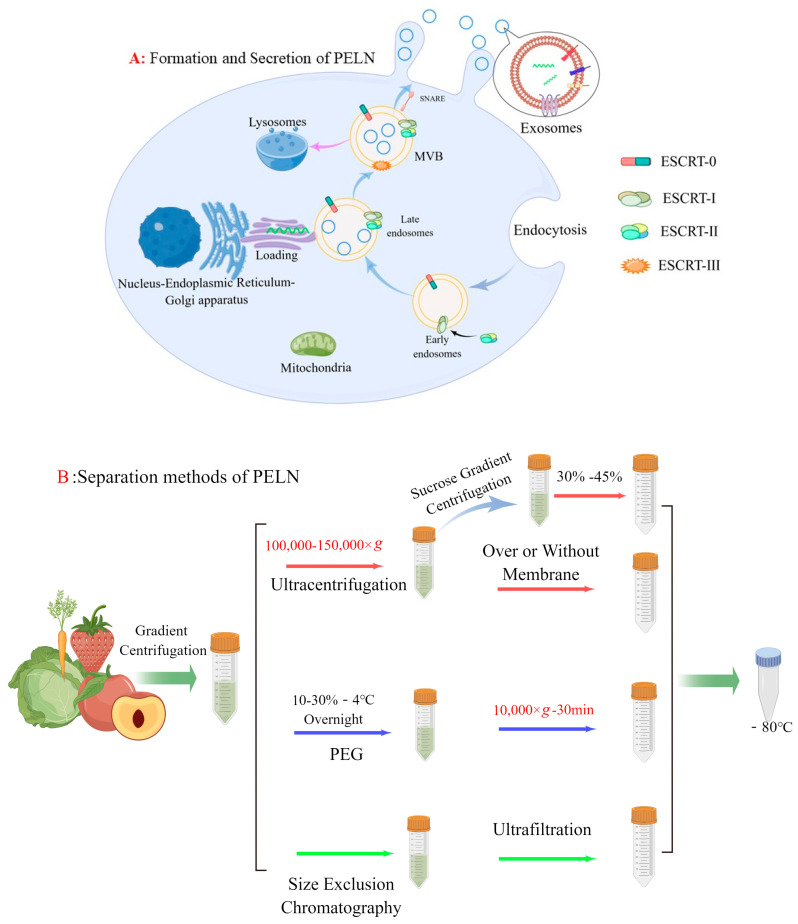
(**A**) Formation and secretion of PELN. (**B**) PELN separation methods (by Figdraw).

**Figure 2 foods-13-00712-f002:**
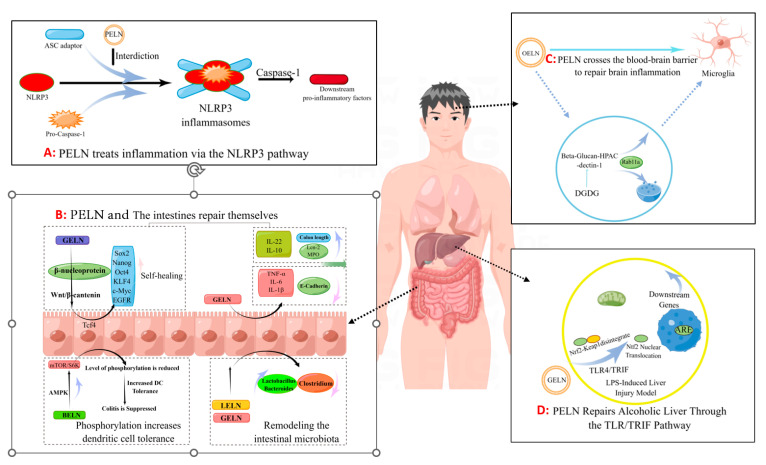
The pathway by which PELN regulates inflammation (by Figdraw).

**Figure 3 foods-13-00712-f003:**
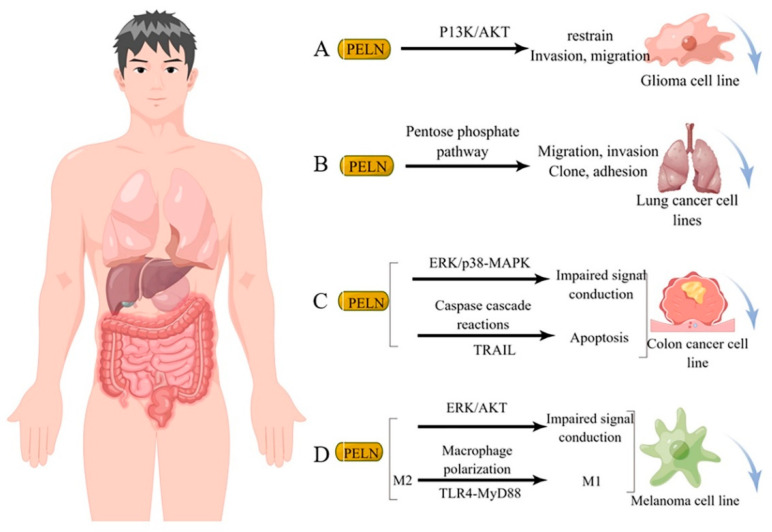
Potential mechanism underlying tumor suppression by PELN. (**A**): PELN and glioma; (**B**): PELN and lung cancer; (**C**): PELN and colon cancer; (**D**): PELN and melanoma. (by Figdraw).

**Table 1 foods-13-00712-t001:** Biological functional mechanism of action of PELN.

Sources	Isolation	Model	Omics	Mechanism of Action	Ref.
Orange juice	UC	HFHSD	Lipids, Metabolism	TLR4—gut repair	[64]
Ginger	SDC	DSS	Lipids, Metabolism, Proteins	Gut epithelial cell proliferation, mitigating damage factors—gut repair	[6]
Broccoli	UC	DSS		AMPK-DCIncreased cell tolerance—colitis	[10]
Tea	SDC	DSS	Lipids, Metabolism, Proteins	Homeostasis of the gut microbiota, prevention and treatment of gut inflammation	[71]
Grape	SDC	DSS	Lipids,	Wnt/β-cantenin-TCF4 Transcriptional mechanism—colitis	[9]
Ginger	SDC		Lipids, Proteins RNA	Wnt/TCF4 signaling pathway—gut homeostasis	[72]
Grapefruit					
Carrot					
Ginger	SDC		Lipids, Proteins, miRNA	Lactobacillus rhamnosus—gut microbiota, gut inflammation	[14]
Ginger	SDC			Inhibits the activation of NLRP3 inflammasomes	[73]
Blueberry	UC		Metabolism, Proteins, miRNA	Inhibits oxidative stress of TNF-α	[74]
Lemon	SDC			Molecular pectin-enzyme R-NaseP-msp1, msp3-bile resistance	[75]
Garlic	SDC	HFD		c-Myc-regulates the cGAS/STING pathway—inflammation of the brain IDO1-AHR signaling pathway—insulin resistance	[76]
Honey	SEC		miRNA	Inhibit the activation of the NLRP inflammatory plateau	[8]
Carrot	SEC		Proteins	Nrf-2 antioxidant mechanism	[77]
Oats	ODC	Alcohol Brain		HPCA (hippocampal calcin)/Rab11a/dectin-1-brain inflammation	[7]
Lemon	UC	LPS	Metabolism, Proteins	Nuclear translocation of NF-κB—inhibits ERK1-2/NF-κB-anti-inflammatory	[11]
Shitake	UC	GalN/LPS	Proteins	Inhibits NLRP3 inflammasome—liver damage	[78]
Garlic	UC	LPS	Proteins	LPS-induced inflammation	[79]
Ginger	SDC	Alcohol Liver	Lipids, Metabolism	TLR4/TRIF-activates Nrf2 nuclear translocation—alcoholic liver	[80]
Ginger	SDC	Alcohol Brain	miRNA	miRNAs regulate viral protein expression—lung inflammation	[81]
Broccoli	SDC		Metabolism, Proteins	Antiproliferative and antioxidant	[82]
Grapefruit	UC	A375	Metabolism	P13K/AKT and MAPK/ERK, inhibit tumor growth	[12]
Ginseng	SDC	Macrophages	Proteins, Lipids, RNA	TLR-4 and MyD88-M2 to M1 polarization-apoptosis	[18]
Lemon	SDC	CML-Tumor	Proteins	Induction of TRAIL-mediated tumor growth	[11]
Lemon	SDC		Proteins	Phosphorylation-antitumor of ACACA-ERK1/2 and P38-MAPK	[83]
Lemon	ELD	Gastric Cancer		Stage S-cell cycle arrest and apoptosis in gastric cancer cells	[26]
Momordica charantia	SDC			p-AKT/AKT and p-PI3K/PI3K-glioma	[84]
Garlic	ATP	Tumor Cells		S-phase cell cycle arrest-caspase-mediated apoptosis-anticancer	[85]
Ginseng	UC	Tumor Cells	Proteins	Pentose phosphate pathway activity inhibits epithelial mesenchymal transformation in lung cancer cells	[86]
Wheat	Kits		Proteins	Wound healing potential	[15]
Grapefruit	ATP		Proteins, miRNA	Upregulation of wound healing gene expression	[87]
Momordica charantia	SDC		Proteins	AKT and ERK—mitochondrial dysfunction—cardioprotection	[13]
Coffee	SEC	Liver Tumors		Liver fibrosis in chronic liver disease	[88]
Portulaca oleracea	UC ATP	Cornecocytes		Upregulation of gene expression that builds a skin barrier and prevents water loss	[89]
Green tea					
Ginseng					
Medlar	UC	HaCaT	miRNA	miR-CM1 inhibits Mical2 and inhibits ultraviolet ray-induced skin aging	[90]
Watermelon	UC	Caco-2	Proteins	Alters gut communication with distant tissues—improves placental function and reduces FGR	[91]

UC: ultracentrifugation; SDC: sucrose density gradient centrifugation; SEC: Size exclusion chromatography; ODC: OptiPrep density gradient centrifugation; ATP: waterborne two-phase system; HDHSD: a model of a high-fat, high-sucrose diet; DSS: model of induced colitis; HFD: High-fat diet;ELD: electrophoretic technique with 300 kDa cut-off dialysis bag.

**Table 2 foods-13-00712-t002:** Comparative analysis of drug loading systems for PELN.

Source	Grooming Process	Target	Loads	Function	Ref.
Grapefruit		Gut macrophages	Anti-inflammatory methotrexate (MTX)	Reduces the toxicity of MTX and improves its therapeutic efficacy for DSS-induced colitis in mice	[69]
Grapefruit	Folic acid (FA)	Gut	Biotin siRNA, JSI-124		[53]
Grapefruit	FA-PEI-GNVs	Brain-GL-6 tumors	miRNA17	Delayed brain tumor growth	[9]
Cherry			Nucleic acid		[104]
Cabbage			Therapeutic drugs		[16]
Lemon	Heparin-cRGD-EVs-doxorubicin	Liver, Spleen, kidney, and tumors	Doxorubicin	Multidrug resistance in cancer	[102]

## Data Availability

The original contributions presented in the study are included in the article, further inquiries can be directed to the corresponding author.

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
