# Peer review of "Exploring the Potential of Plant-Derived Exosome-like Nanovesicle as Functional Food Components for Human Health: A Review"

_foods, 2024, doi:10.3390/foods13050712_

Round 1

Reviewer 1 Report

Comments and Suggestions for Authors

The provided manuscript focuses on extracellular vesicles (EVs) and, more specifically, plant-derived exosomes. It highlights the characteristics of these particles, such as their membrane structure, their diversity in sources, and their potential role in food nutrition research, herbal activity evaluation, and other fields. It also addresses the challenges in extracting exosomes from plants and highlights the potential applications of these nanocarriers, underlining their ability to transport active molecules to specific tissues. Although it highlights the growing interest in investigating the physiological functions of these vesicles, it points out unresolved questions about the selective incorporation of active substances and the stability of the processing of plant exosomes. The text proposes a comprehensive overview of the topic and suggests future research directions.

It is a well-done review. However, some corrections need to be made

Include abstract and keywords

Respect and unify the format of citations and references according to the journal guidelines

Improve the quality and dimensions of the figure legends; remember that every graph or figure must contain complete information to explain itself without reading the text.

Check the font size of titles and subtitles

Table 1 is completely incorrectly dimensioned; please adjust it

Author Response

The provided manuscript focuses on extracellular vesicles (EVs) and, more specifically, plant-derived exosomes. It highlights the characteristics of these particles, such as their membrane structure, their diversity in sources, and their potential role in food nutrition research, herbal activity evaluation, and other fields. It also addresses the challenges in extracting exosomes from plants and highlights the potential applications of these nanocarriers, underlining their ability to transport active molecules to specific tissues. Although it highlights the growing interest in investigating the physiological functions of these vesicles, it points out unresolved questions about the selective incorporation of active substances and the stability of the processing of plant exosomes. The text proposes a comprehensive overview of the topic and suggests future research directions. It is a well-done review.

However, some corrections need to be made Include abstract and keywords Respect and unify the format of citations and references according to the journal guidelines

Response: Thank you so much for recognizing this work. We have taken your suggestions into careful consideration and made revisions to the abstract, keywords, and citation format in accordance with the journal guidelines.

Improve the quality and dimensions of the figure legends; remember that every graph or figure must contain complete information to explain itself without reading the text.

Response: Thank you for your valuable advice. We have made adjustments to enhance the image quality in the article. Additionally, we have supplemented the image captions with extra information, ensuring understanding without relying solely on textual content.

 Check the font size of titles and subtitles Table 1 is completely incorrectly dimensioned; please adjust it

Response:Thanks for your comment. The font size has been checked and revised according to your comment.

Reviewer 2 Report

Comments and Suggestions for Authors

MANUSCRIPT: 2789778

TITLE: Exploring the Potential of Plant-derived Exosome-like Nano-vesicle as Functional Food Components for Human Health

The manuscript 2789778 “Exploring the Potential of Plant-derived Exosome-like Nano-vesicle as Functional Food Components for Human Health” presents an interesting study but it is not clear if the authors present a review article or a research article

The idea of the manuscript entitled “Exploring the Potential of Plant-derived Exosome-like Nano-vesicle as Functional Food Components for Human Health” is quite interesting. After a careful review of the manuscript, I found that the manuscript is presented as a compendium of results without a proper discussion that builds a general idea.

The work is more a compilation than a review.

The authors need to work hard in order to give more sense to the manuscript and to connect the sections.

The presentation should be more attractive to the reader. In my opinion, the manuscript needs a reorganization and should be re-structured (namely section 5.4. Absorption and distribution of PELN in the body, should be presented before the sections on the biological effects of PELNs) and complemented with more scientific/experimental information from literature data.

In my opinion, the manuscript fits better in journals in the biomedical sciences than in journals in the food sciences and I recommend that authors, taking into account the observations indicated above, reformulate their manuscript and submit it to Journals in the field of Health or Biomedical Sciences.

Author Response

Reviewer: 2

The manuscript 2789778 “Exploring the Potential of Plant-derived Exosome-like Nano-vesicle as Functional Food Components for Human Health” presents an interesting study but it is not clear if the authors present a review article or a research article The idea of the manuscript entitled “Exploring the Potential of Plant-derived Exosome-like Nano-vesicle as Functional Food Components for Human Health” is quite interesting. After a careful review of the manuscript, I found that the manuscript is presented as a compendium of results without a proper discussion that builds a general idea. The work is more a compilation than a review. The authors need to work hard in order to give more sense to the manuscript and to connect the sections. The presentation should be more attractive to the reader.

Response: Thank you very much for your suggestions. We have further strengthened the discussion of our paper and developed our unique insights, thereby enhancing the quality of the paper. For more details, please refer to the highlighted sections in the article.

 In my opinion, the manuscript needs a reorganization and should be re-structured (namely section 5.4. Absorption and distribution of PELN in the body, should be presented before the sections on the biological effects of PELNs) and complemented with more scientific/experimental information from literature data.

Response: Thank you so much for your valuable comments. We strongly believe that the stability, safety, and cellular absorption of PELN are crucial factors in determining its potential application value. Therefore, we have included them all in the same chapter.

In my opinion, the manuscript fits better in journals in the biomedical sciences than in journals in the food sciences and I recommend that authors, taking into account the observations indicated above, reformulate their manuscript and submit it to Journals in the field of Health or Biomedical Sciences.

Response: Thank you very much for your comments. Most of the cases discussed in this article come from edible plants. PELN, as a potential nutrient in edible plants, will provide a new perspective for food nutrition evaluation and functional food development, so we believe that this review should be published in food journals.

Reviewer 3 Report

Comments and Suggestions for Authors

As it is clearly indicated in your review, PELN is arriving with numerous components / non-resolved issues which are inhibiting its development. And it is also described the potential synergistically functions that active substances may play (i.e., secondary metabolites, miRNA) once are incorporated into PELN. Applications of PELN and physiological functions concerns are some of the objectives contain in your research work.

It seems to have designed a good strategy for this investigation:

Isolation and purification of PELN (Formation / Extraction / PELN production/ etc. // PELN composition (Proteins and lipids / Nucleic acids including different RNA's / Phytochemicals // PELN - biological functions (A big table on on mechanism of biological functions related to the action of PELN / Another second table on drug loading systems for PELN / Gut microbiota changes by the effects by the delivery capabilities of PELN / Inflammation improvements / Effects on tumors // Potentialities in the application of PELN with varios components. 

Finally, conclusions which have been termed Prospects. It has been described a high number of prospects and possibilities based on the remarkable potentialities of PELN. Perhaps authos could be more cautious in this section because the wide range of options should be, at this stage, not as wide; but this reviewer leaves that decision to authors.

This is an excellent piece of work produced by authors. And it is suggested to publish as it is but after including a minor reduction of options described by authors.  

Author Response

Reviewer: 3

As it is clearly indicated in your review, PELN is arriving with numerous components / non-resolved issues which are inhibiting its development. And it is also described the potential synergistically functions that active substances may play (i.e., secondary metabolites, miRNA) once are incorporated into PELN. Applications of PELN and physiological functions concerns are some of the objectives contain in your research work.It seems to have designed a good strategy for this investigation:Isolation and purification of PELN (Formation / Extraction / PELN production/ etc. // PELN composition (Proteins and lipids / Nucleic acids including different RNA's / Phytochemicals // PELN - biological functions (A big table on on mechanism of biological functions related to the action of PELN / Another second table on drug loading systems for PELN / Gut microbiota changes by the effects by the delivery capabilities of PELN / Inflammation improvements / Effects on tumors // Potentialities in the application of PELN with varios components.

Finally, conclusions which have been termed Prospects. It has been described a high number of prospects and possibilities based on the remarkable potentialities of PELN. Perhaps authos could be more cautious in this section because the wide range of options should be, at this stage, not as wide; but this reviewer leaves that decision to authors.

Response: Thank you so much for your valuable advice. Taking into account the feedback from reviewers, we have made some modifications to this particular section.

This is an excellent piece of work produced by authors. It is suggested to publish as it is but after including a minor reduction of options described by the authors. 

Response: Thank you very much for your recognition of this work and your valuable suggestions. According to your suggestions, we have streamlined part of the paper.

Reviewer 4 Report

Comments and Suggestions for Authors

After careful review I found the manuscript 'Exploring the Potential of Plant-derived Exosome-like Nanovesicle as Functional Food Components for Human Health' thoroughly investigates plant-derived exosome-like nanovesicles (PELNs), emphasizing their health benefits and roles in functional foods. It encompasses the formation, isolation, purification, characterization, and biological functions of PELNs, alongside discussing the field's complexities and future research opportunities. However, there are some areas of concern that need to be addressed prior to considering the manuscript for publication.

Comments

1.      Structure and Clarity: The manuscript is well-organized, but certain sections could be streamlined for clarity. I recommend a more concise presentation of information to enhance readability, especially in the sections detailing the formation and purification of PELNs.

2.      Comparative Analysis: The comprehensive review of PELNs is commendable. However, it would be beneficial to include a detailed comparative analysis highlighting the advantages and disadvantages of PELNs in relation to other nanovesicles and functional food components, aiding in contextualizing PELNs within the broader field.

3.      Practical Applications: The manuscript thoroughly discusses theoretical aspects but would benefit significantly from incorporating more examples of PELNs' practical applications. Specifically, elaborating on how PELNs can be integrated into food products would provide valuable insights into their real-world applicability.

4.      Future Research Directions: The discussion on future research is insightful. Expanding this section to include more concrete recommendations for experimental designs and potential interdisciplinary collaborations could offer a clearer roadmap for future studies in this field.

5.      Graphics and Illustrations: The manuscript could be greatly enhanced by adding more figures or infographics. These should specifically aim to demystify complex concepts and processes related to PELNs. For instance, visual representations of the isolation and purification processes, as well as the mechanism of action of PELNs in a biological context, would be particularly beneficial.

Author Response

Reviewer: 4

After careful review I found the manuscript 'Exploring the Potential of Plant-derived Exosome-like Nanovesicle as Functional Food Components for Human Health' thoroughly investigates plant-derived exosome-like nanovesicles (PELNs), emphasizing their health benefits and roles in functional foods. It encompasses the formation, isolation, purification, characterization, and biological functions of PELNs, alongside discussing the field's complexities and future research opportunities. However, there are some areas of concern that need to be addressed prior to considering the manuscript for publication.

1.Structure and Clarity: The manuscript is well-organized, but certain sections could be streamlined for clarity. I recommend a more concise presentation of information to enhance readability, especially in the sections detailing the formation and purification of PELNs.

Response: Thank you so much for your valuable advice. In response to your question about simplifying the structure, we simplified the purification part of PELN. We omitted the specific processes of separation methods such as ultrafiltration and PEG, as these have been reported in many literature sources. Additionally, we retained our discussion on formation and purification because these were issues we encountered during the extraction process. This can provide readers with certain solutions when encountering similar issues in practical experimental processes.

  1. Comparative Analysis: The comprehensive review of PELNs is commendable. However, it would be beneficial to include a detailed comparative analysis highlighting the advantages and disadvantages of PELNs in relation to other nanovesicles and functional food components, aiding in contextualizing PELNs within the broader field.

Response: Thank you very much for your comments. According to your comments, we have described the advantages of PELN over other nanovesicles, such as variety, safety, etc.

  1. Practical Applications: The manuscript thoroughly discusses theoretical aspects but would benefit significantly from incorporating more examples of PELNs' practical applications. Specifically, elaborating on how PELNs can be integrated into food products would provide valuable insights into their real-world applicability.

Response: Thank you for your valuable advice. We have made diligent efforts to identify practical applications, but the majority of PELN research remains in the theoretical stage. Consequently, we can only include a limited number of PELN application cases, such as the clinical application of GELN, in our paper.

4.Future Research Directions: The discussion on future research is insightful. Expanding this section to include more concrete recommendations for experimental designs and potential interdisciplinary collaborations could offer a clearer roadmap for future studies in this field.

Response: Thank you very much for your suggestion. we have provided a more specific discussion in the Prospects section of PELN. For example, we have discussed more intelligent and efficient extraction methods, as well as how to conduct interdisciplinary research, such as combining with molecular biology to explore the molecular mechanisms of PELN's function, in order to provide precise nutritional assistance.

5. Graphics and Illustrations: The manuscript could be greatly enhanced by adding more figures or infographics. These should specifically aim to demystify complex concepts and processes related to PELNs. For instance, visual representations of the isolation and purification processes, as well as the mechanism of action of PELNs in a biological context, would be particularly beneficial.

Response: Thank you very much for your advice. The extraction process of PELN was shown in Figure 2. As for the mechanism of action of PELN, we are represented by Figures 3 and 4.

Round 2

Reviewer 1 Report

Comments and Suggestions for Authors

The images are of poor quality; some words cannot be read, especially Fig 1B; it still has a visible watermark that calls into question the origin of the images.

Author Response

The images are of poor quality; some words cannot be read, especially Fig 1B; it still has a visible watermark that calls into question the origin of the images.

Response: Thank you for your question. We greatly appreciate it. In this revision, we have made significant improvements to the quality of several figures, particularly Figure 2, which has significantly enhanced readability. Additionally, we have rectified the quality issue with Figure 1B and clearly indicated the tools used in the article. Furthermore, we have obtained proper licensing for these tools.

Reviewer 2 Report

Comments and Suggestions for Authors

MANUSCRIPT: 2789778

TITLE: Exploring the Potential of Plant-derived Exosome-like Nano-vesicle as Functional Food Components for Human Health

Revised manuscript 2789778 “Exploring the Potential of Plant-derived Exosome-like Nano-vesicle as Functional Food Components for Human Health”.

The authors present the new manuscript reformulated according to almost all the reviewers' recommendations, however, some questions still remain to be clarified or resolved

Regarding the manuscript, I only have small questions for the authors to consider:

1. Authors are recommended to identify the manuscript as a review article or a research article

2. Figures 1, 2 and 3 – It is recommended to pay attention to the figures presented. Please note the origin of figures if it was not created by the authors of the manuscript. Note Figures taken directly from the literature without alteration should include the original reference source in the legend and should say “Taken From Ref. XX with permission of (publisher)” or “Published with permission of XX publisher” and cite the publication. Modified figures should say “Adopted from Ref. XX” and nothing else is needed.

3. Subsection 5.4. Absorption and distribution of PELN in the body, should be presented in section 4. Biological effects of PELNs.

Author Response

The authors present the new manuscript reformulated according to almost all the reviewers' recommendations, however, some questions still remain to be clarified or resolved

Regarding the manuscript, I only have small questions for the authors to consider:

  1. Authors are recommended to identify the manuscript as a review article or a research article

Response: Thank you so much for your valuable feedback. In this revision, we have added the word "a review" to the title of this article.

  1. Figures 1, 2 and 3 – It is recommended to pay attention to the figures presented. Please note the origin of figures if it was not created by the authors of the manuscript. Note Figures taken directly from the literature without alteration should include the original reference source in the legend and should say “Taken From Ref. XX with permission of (publisher)” or “Published with permission of XX publisher” and cite the publication. Modified figures should say “Adopted from Ref. XX” and nothing else is needed.

Response: Thank you so much for your suggestion. The illustrations in this article were created by the author using the free online platform Figdraw (https://www.figdraw.com/), and we have obtained permission from the website to use them. The image annotation also mentions the drawing tool used.

3. Subsection 5.4. Absorption and distribution of PELN in the body, should be presented in section 4. Biological effects of PELNs.

Response: Thank you very much for your valuable comments. We have considered your suggestion and have incorporated subsection 5.4 in Section 4 
